# Nature or Nurture? Role of the Bone Marrow Microenvironment in the Genesis and Maintenance of Myelodysplastic Syndromes

**DOI:** 10.3390/cancers13164116

**Published:** 2021-08-16

**Authors:** Syed A. Mian, Dominique Bonnet

**Affiliations:** Haematopoietic Stem Cell Laboratory, The Francis Crick Institute, London NW1 1AT, UK; Syed.mian@crick.ac.uk

**Keywords:** myelodysplasia syndromes, hematopoiesis, haematopoietic stem cells, clonal haematopoiesis of indeterminate potential (ChIP), inflammaging, bone marrow microenvironment, mesenchymal stomal cells, endothelial cells, immune-bone marrow microenvironment

## Abstract

**Simple Summary:**

Myelodysplastic syndromes (MDS) like many other blood cancers is a disease of the bone marrow, in which the spongy part of the bone is not able to produce enough healthy blood cells. MDS is primarily a disease of the elderly, but it can affect people at a younger age as well. The disease arises as a result of a combination of complex processes that is thought to be primarily driven by accumulation of genetic mutations in the stem cells. However, there is an increasing evidence implicating the bone marrow environment as a fertile milieu where these mutated stem cells thrive and give rise to the disease. In this review, we have discussed the role of blood stem cells and how other cell types in the bone marrow environment interact with each other, therefore contributing to MDS. In addition, we discuss the therapeutic strategies that can be exploited to treat MDS.

**Abstract:**

Myelodysplastic syndrome (MDS) are clonal haematopoietic stem cell (HSC) disorders driven by a complex combination(s) of changes within the genome that result in heterogeneity in both clinical phenotype and disease outcomes. MDS is among the most common of the haematological cancers and its incidence markedly increases with age. Currently available treatments have limited success, with <5% of patients undergoing allogeneic HSC transplantation, a procedure that offers the only possible cure. Critical contributions of the bone marrow microenvironment to the MDS have recently been investigated. Although the better understanding of the underlying biology, particularly genetics of haematopoietic stem cells, has led to better disease and risk classification; however, the role that the bone marrow microenvironment plays in the development of MDS remains largely unclear. This review provides a comprehensive overview of the latest developments in understanding the aetiology of MDS, particularly focussing on understanding how HSCs and the surrounding immune/non-immune bone marrow niche interacts together.

## 1. Introduction

Myelodysplastic syndrome (MDS) represents a collection of clonal haematopoietic stem cell (HSC) disorders primarily diagnosed in elderly, with a median age of diagnosis at 70 years of age. MDS is driven by a complex combination(s) of genetic changes, that results in heterogeneity in both clinical phenotype as well as the disease outcomes, with a high propensity to develop acute myeloid leukaemia (AML) [1,2]. Significant progress has been made over the last decade in explaining the disease heterogeneity in MDS by the discovery of various somatic gene mutations, particularly in RNA-splicing, chromatin modification, DNA methylation, signal transduction, cohesion regulation and transcription factors, with an average mutation rate of 8 to 12 aberrations per exome [3,4,5]. Some of these gene mutations, for example, TP53 and SF3B1, are slowly being integrated into clinical diagnosis and been shown to have an independent impact on overall survival [6]. Although MDS is typically observed in older adults with the acquisition of somatic gene mutations, it can also present in children and younger adults, albeit at a lower frequency. Interestingly, the aetiology of the inherited MDS is associated with germline predisposition in genes involved in telomere maintenance, DNA repair, biogenesis of ribosomes and cell proliferation. Further detailed reviews on the genetics and clinical implications of various gene mutations in these disorders have been extensively covered elsewhere [7,8,9,10]. Recent studies have suggested that the addition of molecular data to the prognostic scoring systems can improve its predictive applicability in the clinical settings [6,11,12,13], that will not only help in the clinical diagnosis but can also be used for the monitoring of disease progression. Furthermore, there is increasing evidence, recognizing that MDS is, like other cancers, shaped by a combination of recursive rounds of positive clonal selections, where genetic aberrations in HSCs play a central role. Although the HSC clonal dominance is known to be universal in MDS at the time of diagnosis, these genetic changes do not always provide a conspicuous advantage to the malignant cells. Notably, these malignant clones continue to coexist alongside healthy HSCs, which are somehow supressed. Clonal haematopoiesis of indeterminate potential (ChIP), a term recently coined by Steensma et al. [14], described clonal haematopoiesis with MDS/AML-like mutations in individuals with normal blood composition or without any overt haematological malignancies. Although ChIP itself is not considered a malignant stage, at least for now, population-based studies using whole-exome sequencing technology, have revealed ~10-fold increased relative risk of developing myeloid malignancies over several years of follow-up in ChIP cases [15,16,17]. Furthermore, researchers have also observed a previously unreported association of ChIP with autoimmune diseases, such as in osteoarthritis patients [18]. Existence of clonal populations in the bone marrow, typically identified by the expansion of more than one genetically distinct cell populations, is increasingly being viewed as a phenomenon named ‘inflammaging’, the age-associated surge in systemic inflammation (Figure 1). Growing evidence suggests that ChIP itself may reflect certain aspects of the aging process [19,20], particularly in the BM tissue. Furthermore, accumulating evidence suggests that the bone marrow microenvironment (BMME) could be a key mediator that is providing a ‘fertile inflamed’ milieu where interactions between these components result in disease homeostasis. The purpose of this review is to provide an overview of our current understanding of the MDS pathophysiology, particularly summarizing the interactive role of Haematopoietic stem cells and the niche components, during initiation and progression of the disease.

## 2. Bone Marrow Cellular Metropolis and the Hematopoietic Stem Cells 

Like any major metropolitan city in the world, a cell teams with surrounding ‘specialized workers’ to carry out its daily operations in a highly regulated manner. Detailed reviews on the bone marrow hematopoiesis and niche have been extensively covered elsewhere [21,22,23,24]. We will briefly provide a summarized perspective here. Human bone marrow, a cellular metropolis, is a highly vascularized multicellular tissue containing self-renewing HSCs which generate progeny that progressively differentiates into mature myeloid, erythroid and lymphoid cells. These HSCs in the bone marrow are surrounded by a multitude of different cell types, including osteoclasts, osteocytes, adipocytes, sympathetic neurons, non-myelinating Schwann cells, the largest among all endothelial cells (ECs) and mesenchymal stromal cells (MSCs) [22] (Figure 2). These BMME cells form distinctively organized niches (i.e., endosteal niche, perivascular niche, arteriolar niche, central medullary niche), with each of these anatomical regions in the bone having a specialized role in maintaining the quiescence, homing and mobilization of the HSCs [22]. Apart from direct contact-based regulation, these niche cells also secrete key signalling molecules that play essential role in maintaining the HSCs in the bone marrow [25,26,27]. The increasing ability to identify BMME cells along with the core proteins they secrete, which support the resident HSCs under normal homeostatic conditions, has been mainly driven by advancement in the generation of murine models. Furthermore, niche dependent regulation of HSC lineage commitment might also underlie changes to hematopoiesis during the ageing process [28]. Recently, it has been shown that the disruption of IGF1 in the niche cells compromises HSC function and causes the accelerated aging of HSCs [29]. 

Bone marrow niches are also home to immune cells, such as myeloid-derived suppressor cells (MDSCs), regulatory T cells (Tregs), dendritic cells, natural killer cells, monocytes, macrophages, T-cells, B-cells and plasma cells that themselves primarily form the ‘immuno-microenvironment’ (Figure 2). Approximately up to 20% of the total bone marrow cells are lymphocytes, with a T-cell/B-cell ratio of 5:1 [30,31]. Altogether, these Immune cells maintain the steady state of the bone marrow, with an increasing number of studies suggesting that immunological challenges, such as autoimmunity, inflammation, bone marrow injury or infection elicit a broad spectrum of immunological reactions in the bone marrow and, in turn, disrupt the function of HSCs and BM niche cells [32]. For example, Tregs in the bone marrow provide a protective immune-privileged reservoir that directly regulates quiescence, abundance, and repopulating capacity of HSCs [33,34,35]. Interestingly, HSCs upregulate their ‘don’t eat me’ signal receptor, i.e., CD47, once they leave this safe immune-privileged niche environment, and this upregulation protects them from the phagocytic engulfing process by the cells of the innate immune system as they travel through the vasculature system [36]. Furthermore, the depletion of macrophages themselves led to the reduction in HSC-supportive cytokines in the bone marrow endosteum, which ultimately resulted in the mobilization of HSCs into the peripheral blood of the mice [37]. CXCR4 antagonist or granulocyte colony stimulating factor mediated mobilization of HSPCs resulted in a significant loss of bone marrow macrophages [38]. On the other hand, studies have shown that not only immune cells but also HSCs express innate immune receptors, such as Toll-like receptors (TLRs), and the ligation of receptors results in secretion of pro-inflammatory cytokines, cell migration, proliferation and differentiation into myeloid cell lineage [39]. In addition to evading potential attack from the innate immune cells, HSCs can also protect themselves against adaptive immune surveillance by regulating their CD274 receptor protein levels [40,41]. Altogether, these studies define bone marrow not only as a reservoir of HSCs, its progeny and niche cells, but rather as an organ of active complex reactions, where HSCs are capable of adapting the demand signals to hematopoiesis in response to various hemato-immunological signals, and also to maintain themselves in a steady state.

## 3. Initiators of MDS: HSPCs or BMME

### 3.1. Haematopoietic Stem Cells

MDS being an old age-related disease has long been considered to be an oligoclonal HSC disorder (Figure 1 and Figure 2). In line with this, advances in sequencing technologies have deciphered this heterogenous genetic landscape of MDS with aberrations present in genes involved in various pathways (such as RNA-splicing, epigenetic machinery, transcription factors and cell cycle) that play pivotal roles in normal haemopoietic cell development amongst others [3,4,5,13]. Various studies have revealed that some of these genetic abnormalities originate from early HSCs (CD34^+^CD38^−^CD45RA^−^CD90^+^/CD49f^+^), providing the strongest evidence for the existence of Myelodysplastic syndrome-initiating cells (MDS-ICs) [42,43]. In fact, some of these studies have also demonstrated that MDS BM consists of multiple genetically distinct disease subclone(s) that often follow branching multi-clonal and/or ancestral evolutionary paradigm, both of which can result in intra-tumour heterogeneity [4,42,43]. This intra-tumour heterogeneity has long been suggested as a common feature in both acute leukaemia and solid tumours; however, with a more complex picture in these disorders. Recent studies have evaluated the clonal relationships among primary and metastatic cancers [44,45,46,47]. These reports demonstrated that seeding metastases need few, if any, additional driver lesions beyond those found in the primary malignant cells. Notably, initiating genetic insults that form the trunk of the clonal haematopoiesis seem to get an ‘additive’ effect by certain branching genetic mutation(s), that occur later during the tumour evolution which in-turn contributes to the maintenance, progression and subclonal diversification [42,48]. This fits well with the Darwinian Theory that essentially dictates how the evolution can create systemic biodiversity and this in-turn makes an entire ecosystem more stable. Interestingly, ‘potent’ mutations in classic oncogenes such as TP53 often associated with worst overall survival, in general, are also detected in healthy individuals with no obvious clinical disease. On the other hand, mutations in splicing factors, such as SF3B1, that are also part of ChIP mutation spectrum, have a good prognostic significance in MDS-RS when present alone [5,49]; however, this does not hold true in CLL patients, where these mutations are strongly associated with aggressive disease [50]. This new emerging knowledge raises a question against the traditional concept of clonal dominance (or advantage) with either branching or sequential mutational models that suggest only a ‘potent’ driver mutation(s) is needed to initiate (or transform) the disease. Even if there are beneficial effects of clonal dominance suggested to be acquired by HSCs, there is growing evidence that hints towards the preceding diversity in the form of both background genomic profile (e.g., single nucleotide polymorphisms, genomic loci), epigenetic elements and nongenetic factors that aid in the selection pressure only upon continuous environmental perturbations [51,52,53,54,55,56,57,58,59,60]. Association of a genome-wide set of germline genetic variants identified three genetic loci that were associated with ChIP status, including one locus at TET2 that was African ancestry specific [58]. Another interesting dataset coming from the genomic analysis of MDS patients among atomic bomb survivors in Nagasaki has revealed that the genetic aberration detected in such cases differed significantly from those in de novo or therapy-related MDS [61]. A recent study that elegantly demonstrated such phenomenon sequenced stem cells from the liver/intestine/colon and reported that aberrations accumulated in adult stem cells at a rate of approximately 20 to 40 mutations per year [62]. The mutation rate differed significantly among cells, tissues, and people, and these variations primarily reflected the intrinsic variability in DNA damage and environmental elements (for example, tobacco exposure in bronchial epithelial and lung cells and microbiome in the gut) [63,64,65,66]. In the case of bone marrow tissue, the degree to which such environmental factors initiate nongenetic changes that are eventually replaced with genetic changes, or simply expose a reservoir of existing rare genetic subpopulations, remains completely unresolved.

### 3.2. Mesenchymal Stomal Cells

Over the years, several studies have found that the alteration in the BMME can lead to haematological cancers, especially the establishment of a premetastatic niche that supports the growth and dissemination of clonal neoplastic cells [67]. Most of the work on the role of BMME has been done on leukaemia’s and solid cancers; however, the role that niche plays in the development of MDS is still controversial. Early reports focusing on histological analysis of the patient BM trephines have indicated the disruption of the BM architecture as a common occurrence in MDS [68,69]. This was confirmed by subsequent reports that pointed towards BM-derived stromal cells (i.e., MSCs), as a major component of this disrupted architecture [70,71,72,73]. The clonal origins of MDS MSCs have always been questioned. There are contradictory reports about the presence of ‘MDS related-gene’ mutations or chromosomal abnormalities in MDS MSCs [74,75,76,77,78,79]. Several observations relating to MSCs in patients with MDS, such as altered expression of adhesion proteins, PI3K/AKT signalling, WNT/β-catenin signalling have implied that these cells have animportant role in sustaining the MDS disease phenotype (Figure 2) [71,72,73,74,78,80,81]. MDS patient derived MSCs seems to have reduced clonogenicity, increased senescence and also defects in adipogenic as well as osteogenic differentiation potential, especially in high-risk categories such as refractory anaemia with excess blasts [71,74,78,82,83]. Focal adhesion kinase that regulates cell signalling networks including survival, proliferation, differentiation, mobility, and adhesion, is dysregulated in MDS MSCs and has been corelated with increased senescence and dysfunctional differentiation [72,83]. Interestingly, the overexpression and overactivation of FAK has been associated with tumour aggressiveness [84]. The first experimental evidence that a specific distinct stromal cell type, in this case an osteolineage cell, can initiate BM failure, comes from a study where the disruption (via osterix promoter) of endoribonuclease Dicer1 in immature oestoprogenitors led to myelodysplasia-like syndrome. Interestingly, when the deletion of Dicer1 was induced in mature osteolineage cells through the use of the osteocalcin promoter, no malignancy was observed, demonstrating that the ‘cell type specificity’ in the genetic lesion is essential [85]. However, the role of DICER1 in the MDS pathophysiology has recently been questioned by a report where no evidence was found between the germline DICER1 alterations and onset of MDS disease [86]. Studies have reported reduced expression of DICER1 in MDS MSC [85,87]; therefore, further studies are needed to study the role of DICER1 in MDS disease. Transgenic mouse models, such as NUP98/HOXD13 (NHD13), that develop MDS-like features [88,89], have been reported to have reduced osteoclasts, and increased the amount of nonmineralized bones in the presence of an increased number of osteoblasts and a defective erythropoiesis. This was accompanied by increased fibroblast growth factor-23 (FGF-23) serum levels, and a phosphaturic hormone that inhibits bone mineralization and erythropoiesis. Interestingly, this observation was later validated in primary patients’ samples with MDS [90]. Furthermore, critical contributions of the BMME to the MDS have further been suggested by the inability of human MDS stem cells to propagate in a cell-autonomous manner. Although the co-injection of human MSCs along with MDS CD34+ HSPCs was initially suggested to support MDS engraftment in patient-derived xenograft (PDX) models [91], subsequent reports did not show any beneficial effect of MSCs in PDX models [92,93]. Notably, the requirement of humanized niches for the establishment and maintenance of MDS clonal architecture was recently confirmed in a study in which the role of human MSCs was demonstrated to be crucial. MDS stem cells were able to engraft not only in the autologous humanized ectopic niches, but also in allogenic healthy humanized ectopic niches in PDX xenograft models. Interestingly, these MDS stem cells were able to migrate and were home to other ectopic niches that were pre-seeded with human MSCs. Contrary to healthy donor stem cells, no migration of MDS stem cells was observed with the other murine hematopoietic tissues, or even the ectopic niches seeded with murine MSCs [94]. This preferential migratory behaviour of MDS stem cells has also been reported in other more aggressive haematological cancers, such as AML and ALL [95,96]. Interestingly, HSCs migration is not only associated with malignant conditions, but has also been reported in healthy adults, where the HSCs exit into circulation in order to fill empty bone marrow hematopoietic bone marrow niches [97,98]. Further studies are needed to understand the dynamics of such migration and if this is associated with normal HSC behaviour or points to a more aggressive MDS-IC intrinsic properties or is driven by niche factors. Clinically, donor cell leukaemia, although rare, is the best example of niche-driven disease, where leukaemia originates from engrafted donor cells following allogeneic HSC transplantation [99]. Several studies have also provided evidence for leukemic chemo-resistance mediated by the microenvironment, including a recent report demonstrating protection from apoptosis provided by specific endosteal niches within the mouse BM [100]. In the context of MDS, MDS-MSCs seems to impair the growth and function of healthy hematopoietic stem and progenitor cells (HSPCs), and this defect was sustained autonomously in HSPCs following exposure to MDS MSCs (Figure 2). Co-cultured healthy HSPCs (with MDS MSCs) resulted in a failure for these active HSPCs to engraftment across primary and secondary xenograft recipients [74]. This observation raises the question of whether MDS-MSCs themselves are dysfunctional or if this is a secondary effect induced by the MDS HSCs, and this disease memory not only lasts for long term but can also be transmitted to healthy donor HSCs. Interestingly, just as the BMME may have an influence on the leukemic cells, leukaemia can also remodel the micro-environment for its advantage, therefore suggesting a bi-directional communication [101,102]. Some of the best-studied examples of BM remodelling in haematological malignancies are AML, multiple myeloma (MM), Chronic myelogenous leukemia and T-acute lymphoblastic leukaemia [103,104,105]. The remodelling of the perivascular niche as a result of the autocrine and paracrine secretion of vascular endothelial growth factor (VEGF) and other factors by leukaemia led to an increased proliferation of microvascular endothelial cells, as well as leukemic blasts in AML [106]. In CML, leukemic cells initiate MSC differentiation into an aberrant pro-fibrotic osteoblastic lineage, that in-turn promotes leukaemia expansion at the expense of normal haematopoiesis [107]. It is tempting to speculate that, similar to leukaemia, MDS-ICs alter the BM micro-environment that provides a nurturing niche for the sustenance of MDS-ICs and suppression of healthy HSCs, and even contributes to the emergence, as well as the evolution, of neoplastic clones. 

### 3.3. Endothelial Cells

Endothelial cells (ECs), a core component of the vascular niche in the BM, have been harder to dissect in detail, partly due to the lack of reliable phenotypic markers identifying endothelial progenitors. Although the role of ECs in MDS pathogenesis is not well studied, there is a growing evidence suggesting the existence of a dysfunctional vascular niche that has the potential to promote MDS initiation and disease progression (Figure 2). The first report to suggest the existence of a dysregulated vascular niche comes from the observed morphological feature later termed as the “abnormal localization of immature precursors” (ALIP), a histopathological hallmark of MDS. The term itself describes the abnormal localization of HSPCs in the BM interstitium, rather than the paratrabecular endosteal niche, as typically observed in healthy BM [68,108,109]. Such a phenotype has recently been described in AML xenograft models, wherein endosteal regions were depleted from human HSCs in the murine BM [110,111]. MDS patients are reported to have increased BM vascular density, which corelates with increased BM myeloblasts and advanced disease subtype [112,113,114]. Angiogenic growth factors, such as VEGF, Ang-1, angiogenin, FGF-β, and HGF, are also significantly elevated in the MDS patients [115,116,117,118]. Furthermore, VEGF dysregulation promotes both paracrine signalling to mediate the remodelling of the BM microenvironmental, and autocrine growth, as well as the proliferation of MDS progenitors. Interestingly, the stimulation of VEGF increased the colony forming ability of primary MDS (high-risk MDS) cells, while the neutralization of VEGF activity produced an inverse colony output [119]. The induction of VEGF increases vascular permeability in sinusoidal ECs which, in turn, stimulates the cell cycle, migration and differentiation of HSPCs, but with myeloid bias and increased apoptosis [120]. These features are a hallmark of MDS haematopoiesis, and at least partly provide a biologic rationale for ALIP and its association with adverse prognosis in high-risk MDS. The molecular defects suggested to be present in MDS endothelial progenitors seem to be intrinsic in nature, since these cells have distinct transcriptomic and epigenomic profiles, enhanced autophagy and lysosomal degradation features and have aberrant HSPC supporting ability [119,121,122]. This observation is further supported by another study that reported circulating ECs (from 40% to 84%) from MDS patients had the same cytogenetic abnormalities as the MDS HSC clone [123]. Further studies are needed to conclusively determine if the BM ECs forming the vascular milieu are themselves dysfunctional and, if any, the role they play in driving myelodysplasia.

### 3.4. Immune-BMME

While there is more evidence suggesting that the BM microenvironment influences malignant hematopoiesis, the mechanism leading to MDS-associated immune suppression is still largely unknown and, in fact, controversial. MDS patient-derived classical monocytes have been shown to have high expression levels of thrombomodulin, a molecule with anti-inflammatory properties (Figure 2) [124]. It remains to be determined whether this increase in the expression level of anti-inflammatory molecules in monocytes is directly driven by an intrinsic transcriptional machinery or via an intermediate external force. Notably, studies have demonstrated that a subgroup of MDS patients has increased the number of classical monocytes and is associated with poor prognosis [125,126,127]. Sarhan et al. in a recent study reported that healthy donor monocytes upon culture with MDS MSCs, acquired phenotypic, metabolic and functional properties of myeloid-derived suppressor cells (MDSCs), and this was accompanied with the suppression of NK cell function, as well as T cell proliferation by ‘MDS MSC educated’ monocytes. This MSC driven mechanism demonstrates that immune suppression can be communicated from the malignant cells to its stroma via an indirect mechanism, which ultimately benefits the MDS clones [128]. 

Macrophages are another cell type that are essential component of the innate immune system and their main function is to ingest as well as degrade abnormal cells (or cellular debris), and drive inflammatory processes. Macrophages are also known to be directly regulating HSCs via CD234/CD82 (to inhibit cell cycle progression), CD169 (promote erythroblast differentiation into reticulocytes and destroy the aging erythrocyte) and VCAM-1 (HSC homing) [129,130,131]. Tumour-associated macrophages (TAMs) play a fundamental role in the pathophysiology of human cancers, in general [132]. However, this association is weak in haematological malignancies, with only recent data suggesting some association of TAMs with poor disease outcomes in blood cancers, such as lymphoma, multiple myeloma and leukemia [133]. There is emerging evidence suggesting that macrophages are also potentially involved in MDS (Figure 2); however, the exact mechanism through which they influence the disease remains to be determined. Studies have suggested that the loss of granulocyte/monocyte progenitor populations in the BM of low-risk MDS, could partly be explained by the increased phagocytosis of these cells by macrophages. This dysregulated process is probably driven by the low-density LRP1 receptor, present on macrophages, acting on the cell surface calreticulin on the target cells [134]. Recently published clinical data have reported a positive correlation between the infiltration of M1-tumor-associated macrophages (TAMs) and overall survival in MDS patients. Interestingly, CD163+ TAM was the main type present in the high risk MDS patients, while this was iNOS+ TAMs in the low risk MDS group [135]. Gene-expression analysis of primary MDS samples has shown altered expression of various genes, including several encoding pro-inflammatory cytokines in the macrophages [136]. 

MDSCs (reviewed in detail elsewhere, Veglia et al. 2021), an immature innate immune cell type, known to accumulate in cancer patients, are effector immunosuppressive cells that contribute to cancer progression [137,138,139]. Notably, MDSCs were reported to be significantly increased in MDS BM (Figure 2) and transgenic MDS mouse models, are distinct from the neoplastic clone and associated with impaired haematopoiesis through a mechanism driven, at least in part, by the interaction of S100A9 with an endogenous ligand for CD33-initiated signalling [140,141,142]. The possible implication of MDSCs in the immune dysregulation associated with MDS and its potential role as biomarkers and therapeutic target has started to attract a particular interest in the field of haemato-oncology [143]. It remains to be determined whether MDSCs are directly implicated or been recruited/educated by other cell types, such as MSCs. More studies are needed to fully dissect the role of MDSCs in MDS disease initiation and progression. 

Recent studies have identified another crucial player in immune regulation, namely FoxP3+ regulatory T cells (Tregs), that are drawing attention for their heterogeneity and noncanonical functions, such as HSC quiescence and engraftment [33,144]. Tregs are high producers of IL10 in the BM and these cells directly regulate MSC function which, in turn, maintain HSCs [145]. One of the emerging pieces of evidence for immune dysregulation in MDS implicates Tregs, which are significantly altered in MDS patients [146,147,148,149], with a decrease in low-risk MDS and an increase in high-risk MDS (Figure 2). The role of Tregs in MDS pathogenesis may potentially explain its strong association with autoimmune disorders and disease progression, considering that dysfunctional and reduced numbers of Tregs can lead to the weakened suppression of excessive immune response, while a high number, along with increased function, can cause the disruption of the immune surveillance machinery against the dysplastic clone(s), therefore allowing for the unhindered proliferation of myeloid blasts. It is still unclear whether this dysfunctional immune regulation is a cause or a consequence of the MDS-ICs. 

## 4. BMME Inflammation: A Friend or a Foe

Altered hematopoiesis and associated immunological changes, predominantly cytokines, in the BM milieu are hallmarks of not only MDS (Figure 2) but also aging. However, a significantly strong linkage between chronic immune stimulation and MDS predisposition has been established in a large cohort of MDS patients [150]. In general, pro-inflammatory cytokines can lead to the activation of ‘emergency transcriptomic switch’ in HSCs which is often associated with the overproduction of myeloid cells and platelet biased; however, this is at the expense of other differentiated lineages in the BM [151,152,153,154,155,156]. This inflammation associated cycling of HSCs can lead to stem cell exhaustion [39,155,157] and may potentially cause the depletion of the HSC pool in the BM. This phenomenon of HSC depletion upon inflammation is often reported in the context of preleukemic clonal changes [158,159,160]. Interestingly, HSC response, initially thought to be a compensatory mechanism to ensure the sufficient supply of differentiated cell types in the blood, is now suggested to be a direct consequence of inflammation stimuli [39,157]. For example, increased production of IL6 by HSPCs, as a result of inflammation, rapidly induces HSC differentiation towards myeloid lineage (referred as myeloid skewing) [157], a phenomenon often associated with age and other myeloid malignancies, such as MDS. In line with this, increasing studies are reporting elevated levels of cytokines, such as TNF, IFNs, IL-6, TGFβ, IL17 and IL-1, play a supporting role in disease maintenance, including MDS and other myeloid malignancies [161,162,163,164,165]. Although experimental evidence suggests that these cytokines can be produced not only by resident macrophages in the BM and circulating lymphocytes but also by stromal niche cells and HSPCs, the exact source remains to be determined. The inflammatory molecules causing constitutive activation of TLR-signalling and subsequent downstream mitogen-activated protein kinase (MAPK) and nuclear factor kappa B (NF-κB) activation have been implicated in the pathogenesis of MDS [166,167,168,169,170,171]. Notably, systematic inflammatory and autoimmune manifestations are also commonly (up to 25%) reported in MDS patients [172,173,174]. Notably, proinflammatory signalling, initiated by TGFβ1 within the BM microenvironment, triggers the molecular alterations and functional inhibition of MSC, which eventually contributes to ineffective haematopoiesis [175]. 

Recent data have suggested that damage-associated molecular pattern (DAMP)-induced inflammation (via oxidized mitochondrial DNA) in the BM of MDS patients can lead to the uncontrolled activation of inflammasome machinery, causing HPSC lytic cell death mediated via pyroptosis, and this can act as a pathogenetic driver of ineffective haematopoiesis in MDS (Figure 2). The levels of oxidized mitochondrial DNA were reported to be highest in MDS compared with other overlapping syndromes and reactive conditions, such as CMML, ChIP, and anaemia [176]. Furthermore, mesenchymal niche can also induce genotoxic stress through p53-S100A8/9-TLR4 inflammatory signalling pathway which directly induces mitochondrial dysfunction, oxidative stress, and the activation of DNA damage responses in HSPCs159. The phenomenon of aberrant inflammasome activation, leading to pyroptosis, an immune mediated cell death program, could at least partially explain the high rate of cell death generally observed in MDS patient BM. Interestingly, S100A9/S100A8 induction can lead to a p53-dependent differentiation defect in erythroblasts, that is reminiscent of del(5q) MDS [177]. In a rheumatoid arthritis mouse model, which is associated with systemic inflammation and myeloid cell bias, inhibiting the inflammatory signal using the interleukin-1 receptor antagonist anakinra led not only to a reduction in inflammation, but also the attenuation of myeloid cell expansion [156]. This study provides a proof of concept for targeting inflammatory signals in order to preserve the balance in HSCs differentiation. Altogether, these studies highlight the importance of HSCs and its progeny, and the way they behave as the first responders to inflammation which has changed our fundamental understanding of HSC biology. It is becoming clear now that inflammation is a necessary protective mechanism by which our body keeps the system in check; however, the question that needs to be addressed is the level and type of inflammation that is good for our body. In fact, when this uncontrolled smouldering inflammatory process continues for a long time, therefore becoming an ‘evil wild fire’—which we often do not notice during the early phases, and which eventually turns to a ‘bad inflammation’—this can induce cell stress, DNA damage and can result in significant BM tissue damage.

## 5. What Options Are There: Targeting the MDS HSPC-BMME Crosstalk

Developing treatment for patients with MDS needs a complex and personalized array of therapeutic approaches. There is a significant lack of progress for treatment options and the only FDA approved treatment that is currently been used in the clinics include the thalidomide analogue lenalidomide and the DNA methyltransferase inhibitors azacytidine and decitabine, and stem cell transplantation [178]. All these drugs are targeting intrinsic cellular vulnerabilities and are limited in terms of their effectiveness. Although erythropoiesis stimulating agents (ESAs) have been instrumental in decreasing the transfusion requirements in low-risk MDS patients, a significant number of patients produce no response or acquire treatment refractoriness. Luspatercept, a recombinant fusion protein that acts as a TGF ligand trap, restoring late-stage erythropoiesis, has recently been tested. Although this drug has yielded promising results by generating transfusion independency [179] in up to 38% of the patients treated with the drug [180,181], more data are needed to fully understand its long-term efficacy. The use of spliceosome inhibitors (H3B-8800), at least in initial stages, provided some excitement; however, a recent report has shown low response rates with some toxicities, such as diarrhoea, nausea, fatigue, and vomiting [182]. Another drug, namely Roxadustat, an oral hypoxia-inducible factor inhibitor is currently being tested in clinical trials involving non-del(5q) low-risk MDS patients with low transfusion burden [183]. This drug has already been approved in China for anaemia related to chronic kidney disease, and works by increasing EPO production, reducing hepcidin and promotes erythroblast maturation [184]. 

Recently, Magrolimab, a monoclonal antibody that specifically interferes with the signal regulatory protein alpha (SIRPa) receptor present on macrophages, which is overexpressed on malignant cells, was tested in a clinical trial alongside azacytidine in high-risk MDS patients. Azacytidine increases the expression of CD47, which leads to the increased binding of magrolimab to CD47, which ultimately results in the increased phagocytosis of tumour cells by macrophages. Notably, this novel combination therapy led to high response rates (complete remission in 42%) and an acceptable toxicity [185] in high-risk MDS patients. Venetoclax, which is already approved by FDA for treating AML is a BH3 mimetic that functions as an inhibitor of BCL2. Various ongoing trials have reported promising results of Venetoclax efficacy in high-risk MDS patients [186,187]. Eprenetapopt (APR-246), a small molecule that selectively induces apoptosis in TP53-mutant cells have provided a hope for high-risk MDS patients with TP53 mutations. Data from phase 2 clinical trials have shown that Eprenetapopt in combination with azacytidine yields high response rates (73%) in TP53-mutated MDS patients, with 50% achieving complete remission and 58% a cytogenic response [188]. There are other novel drugs that are currently being tested in clinical trials for high-risk MDS and are extensively reviewed elsewhere [189]. 

Despite the initial theoretical optimism; however, results from clinical trials targeting BMME (for example, anti-angiogenic drugs) involving AML patients have generated disappointing results. Hedgehog signalling, which is highly active in endothelial cells, has also been targeted in MDS and other leukaemia using small molecule inhibitors with limited success [190]. Finally, there are other novel strategies that are currently being explored to target leukemic BMME such as NOS inhibitors, which ‘normalize’ the abnormal BM vascular permeability associated with leukemia and improves chemotherapy efficacy [110]; however, more data are needed in order for this to be tested in MDS patients. Furthermore, there is growing evidence linking stemness to prognosis and therapy failure in hematological malignancies and, in particular, MDS-IC dependence on the niche cells, therefore identifying and designing drugs that can together target both components (or the cross-talk between the two) along with the elements of MDS-IC stemness, which may be an effective means of eradicating MDS clones. Given the fact that stemness-associated genes are likely to be shared between MDS-ICs and normal stem cells, the successful eradication of MDS-ICs will require understanding to what extent these malignant cells differ from normal stem cells to minimize the impact of therapies on healthy HSCs. The use of single cell RNA/Proteomic data will be essential to delineate this difference in the transcriptional and cell–cell signalling networks that must exist between normal stem cells, MDS-ICs and the BMME cells.

## 6. Future Direction

Although recent developments in MDS HSC genetics have been instrumental in gaining knowledge of disease pathophysiology and has led to the idea of combined targeted therapies, limited understanding of how all of the components in the bone marrow interact remains a major bottleneck. There is an urgent need to understand the role of various specialized niches where these clonal HSCs reside, especially during the early stage of clonal development. The juxtaposition of malignant HSCs carrying multiple mutation(s) within various niches might have an influence on the function as well as the evolution of these clones, resulting in significant variation in the path it takes during the course of the disease. Dissecting the signalling routes that MDS clonal HSCs use to either remodel its environment and directly or indirectly suppress their normal cohabitating HSCs will be of great therapeutic potential. Furthermore, it shall be of future interest to delineate the interaction between the HSC disease and ‘inflammaging’ bone marrow memory. Gaining an understanding of how intrinsic and extrinsic inflammation factors contribute to HSC aging and the development of hematological cancers, in general, might provide novel therapeutic options that can be potentially used alongside existing treatments.

## Figures and Tables

**Figure 1 cancers-13-04116-f001:**
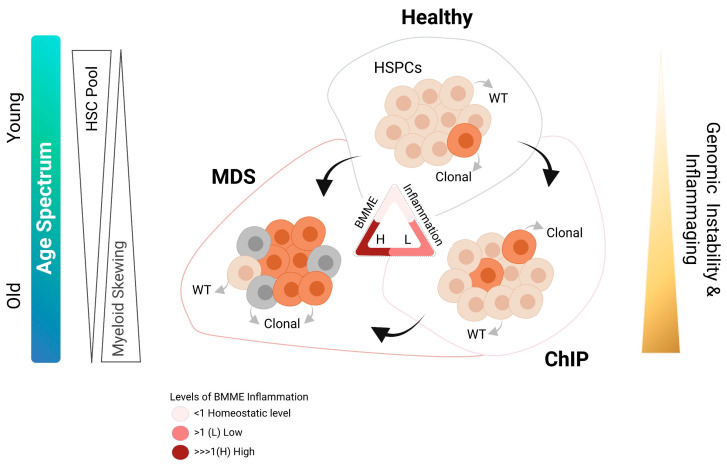
Schematic representation showing possible routes to MDS. Somatic mutations are known to occur in the hematopoietic stem cells (HSCs) during normal DNA replication. The basic premise of the long-standing idea is that a cell that is endowed with an advantageous mutation produces a progeny with increased fitness, which is selected and eventually flourishes. During the early stages of life, bone marrow inflammation is at homeostasis. However, with increasing age, genomic instability, age-related inflammation (Inflammaging) and external environmental cues, these ‘potent’ clones accumulate in the bone marrow, reaching a stage known as clonal hematopoiesis of indeterminate potential (ChIP). Notably, all hematopoiesis is clonally derived; however, malignancy can arise when the hematopoiesis become abnormal or marked by acquisition of additional mutations. Therefore, ChIP can turn to MDS once clinical manifestations of the disease are diagnosed. On the other hand, clonal selection and acquisition of certain mutations (with or without additional external factors) can directly result in switching from healthy stage to MDS. As the bone marrow ages, HSC pool becomes depleted with increasing inflammaging and these cells become skewed towards myeloid lineage. Illustration was created with BioRender.com (accessed on 28 June 2021).

**Figure 2 cancers-13-04116-f002:**
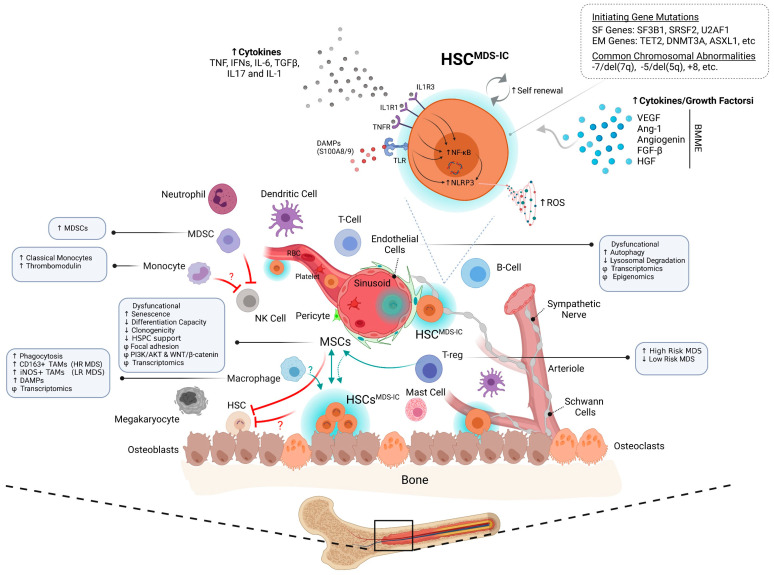
Overview of the MDS bone marrow (BM) cellular metropolis. BM microenvironment is home to multitude of different types of cells including haematopoietic stem cells (HSCs), megakaryocytes, arteriolar and sinusoidal type endothelial cells, osteolineage cells, osteoclasts, osteoblasts, non-myelinating Schwann cells and mesenchymal stromal cells (MSCs). Other cell types that reside in these BM microenvironments are immune-related cells, such as myeloid derived suppressor cells (MDSCs), regulatory T cells (T-regs), dendritic cells, neutrophils, natural killer cells, monocytes and macrophages that primarily form the ‘immuno-microenvironment’. Altogether the interplay between all the components contributes to bone homeostasis as well as the tight regulation of the HSCs (i.e., their homing, quiescence, self-renewal capacity and long-term multilineage repopulation ability), therefore maintaining them in a steady state. MDS-initiating cells (MDS-ICs) with increasing self-renewal capacity are constantly interacting with the cells in the surroundings. MDS-ICs coexist alongside normal wildtype healthy HSCs, which are supressed via an unknown mechanism. MDS-ICs and MSCs are shown to have a bi-directional crosstalk between them. MSCs and endothelial cells themselves have been shown to be dysfunctional. T-regs have been shown to indirectly influence MDS-ICs via MSCs. In MDS bone marrow, a significant increase in the MDSCs and monocytes has been observed. Macrophages seem to have acquired increasing ability for phagocytosis. Abnormal levels of cytokines, growth factors and reactive oxygen species (ROS) are present in the MDS BMME that directly influence the MDS-ICs via various receptors, therefore activating numerous intrinsic pathways (Inflammatory, etc.) in these cells. Φ = Disrupted; ↑ = Increase, ↓ = Decrease, - = deletion. Illustration was created with BioRender.com (accessed on 28 June 2021).

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
