# Peer review of "Nature or Nurture? Role of the Bone Marrow Microenvironment in the Genesis and Maintenance of Myelodysplastic Syndromes"

_cancers, 2021, doi:10.3390/cancers13164116_

Round 1

Reviewer 1 Report

The review conducted by Mian et al. summarized recent advances in myelodysplastic syndromes (MDS), with a focus on hematopoietic microenvironments, and encapsulated the basic immunological mechanism of MDS. Although large parts of the experiments were performed with murine models, the authors also mentioned recent clinical developments and therapeutic strategies. They clearly indicated what aspects of MDS should be clarified. This review is very interesting and informative.

Concern

The figures are excellent but are not referred to in the manuscript. I agree that Figure 2 illustrates an overview of all the steps, but if all graphic elements are mentioned in the paper, readers would easily understand them at each point.

Reviewer 2 Report

This an excellent review article which provides a comprehensive overview of the etiology of MDS,  focusing  on understanding how HSCs and cross talk with the  immune/non-immune bone marrow niche.   The paper is generally well written and structured.  However, in my opinion the
paper has some limitations in regards to some description of  figure and text, and some of the most recent study has not been utilized to its full extent. In several instances I also suggested to cite more relevant and recent
literatures.

  1. Please discuss the clinical syndromes associated with genetic predisposition to myelodysplastic syndrome presenting in children  or younger family MDS. Germline predisposition is increasingly recognized in MDS presenting at older ages as well. Although each individual genetic disorder is rare, as a group, the genetic MDS disorders account for a significant subset of MDS in children and young adults.  The genetic testing plays an important role in the diagnostic evaluation of such clinical setting. This review may provides more information of MDS  associated with genetic predisposition, discusses implications for clinical evaluation and management, and explores scientific insights gleaned from the study of MDS predisposition syndromes. The effects of germline genetic context on the selective pressures driving somatic clonal evolution are explored. Elucidation of the molecular and genetic pathways driving clonal evolution may inform surveillance and risk stratification, and may lead to the development of novel therapeutic strategies.
  2.  The TP53 tumor suppressor gene is the most frequently mutated gene in human cancer, its incidence in MDS/AML is relatively low (5%–20%) and increases with age or in therapy-related MDS/AML. Patients with TP53 mutations have one of the worst prognosis in MDS/AML because their disease is both chemo and immune resistant, as shown by the poor response rate to standard treatments including intensive chemotherapy and hypomethylating agents and the high rate of relapse after allogeneic stem cell transplantation.   The first glimmer of hope came from eprenetapopt (APR-246), which is the first-in-class small molecule that selectively targets TP53 mutated cancer cells.   3. High-throughput DNA sequencing significantly contributed to diagnosis and prognostication in patients with myelodysplastic syndromes.  Thus, large-scale genetic and molecular profiling of multiple target genes is invaluable for subclassification and prognostication in MDS patients. Heterogeneity of tumor population in MDS has now been confirmed to be quite common in MDS and more prominent in advanced WHO subtypes and high-risk prognostic groups.  Therefore, the presence of a hierarchy of mutations was another important finding, which not only advances our understanding of MDS pathogenesis, but also may suggest a clinical relevance in molecular monitoring of disease progression. 
